# Eating Habits during the COVID-19 Lockdown in Italy: The Nutritional and Lifestyle Side Effects of the Pandemic

**DOI:** 10.3390/nu13072279

**Published:** 2021-06-30

**Authors:** Federica Grant, Maria Luisa Scalvedi, Umberto Scognamiglio, Aida Turrini, Laura Rossi

**Affiliations:** CREA Council for Agricultural Research and Economics—Research Centre for Food and Nutrition, Via Ardeatina 546, 00178 Rome, Italy; marialuisa.scalvedi@crea.gov.it (M.L.S.); umberto.scognamiglio@crea.gov.it (U.S.); aida.turrini@crea.gov.it (A.T.)

**Keywords:** SARS-CoV-2, Mediterranean Diet, eating habits, lifestyle, physical activity, food waste, lockdown, Italy

## Abstract

To limit the spread of coronavirus SARS-CoV-2, a nationwide lockdown started in Italy in March 2020. In this unpredictable situation, a cross-sectional study using an online questionnaire was carried out by the Observatory on Food Surplus, Recovery and Waste of CREA Food and Nutrition Centre. The aim of this work was to evaluate how Italian habits changed during this period, the determinants of changes, and the effect on food waste prevention. In a sample of 2678 respondents, 62% showed low Adherence to the Mediterranean Diet (AMD). During lockdown many of participants improved the quality of their diet, increasing their consumption of fruit (24.4%), vegetables (28.5%), legumes (22.1%), nuts (12%), and fish or shellfish (14%). Unfavorable changes were observed with the excessive consumption of sweets or pastries (36.9%) and comfort foods (22.7%), and a lack of physical activity (37.2%). The main novelty of this study was the examination of dietary changes identified by a cluster analysis. Respondents with generally high AMD improved their eating habits, while the habits of the respondents with generally low AMD remained unchanged. In addition, nearly 80% of respondents were sensitive to food waste. The study provides a useful contribution to the debate on nutritional recommendations in case of further lockdown.

## 1. Introduction

In December 2019, the new coronavirus SARS-CoV-2 broke out in China and spread rapidly around the world, leading the WHO to declare it a pandemic in March 2020. As a response, several governments established measures of physical restrictions to contain the virus; in Italy, an emergency law imposing a complete lockdown in the whole country from 8 March 2020 was passed (Legislative Decree 06/2020). Under this law, people were not allowed to leave their homes except for essential activities such as buying food and medicines or for medical reasons. People were encouraged to work from home whenever possible limiting out-of-home activities to essentials. Such a situation had never occurred in Italy as well as in the rest of the world, where similar lockdowns were imposed.

The lockdown period has had significant socio-economic and psychological impacts. In fact, it has been reported that many people suffered from anxiety, depression, and insomnia [1,2]. At the same time, changes in food habits occurred with an increased consumption of unhealthy food [3,4,5] and more time dedicated to cooking [6]. Public health nutrition experts started to emphasize the importance of eating a balanced diet and maintaining an active lifestyle in order to promote adequate dietary habits during the lockdown [7,8,9]. The Mediterranean Diet (MD) principles were recommended [10] given its effectiveness in protecting health [11,12,13,14,15] and promoting an appropriate performance of the immune system [16,17]. Recommendations also included physical exercise, an important determinant of health [18]. Considering the strong connection between nutrition and the immune system, the recommendations stressed the importance of avoiding either malnutrition or overeating since both have a negative impact on the immune response [19,20,21].

In this complex and completely new situation in Italy, the agri-food sector had an important role since consumers immediately reacted by increasing their food stocks. In fact, during the week of 9–15 March 2020, large-scale distribution sales were 30% higher than the same week in 2019. The types of foods purchased during the restriction period resulted in a strong increase in “emergency products” such as long-life milk, flour, eggs, frozen vegetables, and canned meat [22,23]. This situation was also pointed out by the media, which reported a general modification to Italians’ dietary patterns caused by the restrictive measures.

With the purpose of measuring the ongoing phenomena, the research group of the Observatory on Food Surplus, Recycling and Waste of CREA Food and Nutrition Centre [24] conducted a survey on the Italian adult population to investigate eating habits and lifestyle changes, after the containment measures adopted to limit the impact of coronavirus SARS-CoV-2 spreading. When analyzing the results, consideration was also given to whether institutional guidelines [7,8,25] on health, food, and nutrition were properly followed during this period. As innovative aspect, we focused the assessment on the prevention of food waste, an important issue related the pandemic period in Italy [26]. All these aspects are globally aimed to contribute to the debate on the achievement of Sustainable Development Goals [27].

## 2. Materials and Methods

### 2.1. Survey Methodology

A cross-sectional study was carried out on Italian adults (18 years old and above). Participation in the survey was entirely voluntary and anonymous. The study was conducted in agreement with the Declaration of Helsinki [28], and all data were collected and recorded according to the Italian Data Protection Law (Legislative Decree 101/2018) in line with European Commission General Data Protection Regulation (679/2016). The assessment did not involve any invasive procedure nor induced any changes in the participants’ dietary patterns. Therefore, the present web-survey study did not require approval by an ethics committee. Participants were informed about the objective of the research, and they were asked for permission to use and publish the data from the study before starting the data collection. No exclusion criteria were applied.

The questionnaire was distributed through CREA Food and Nutrition Centre institutional website (https://www.crea.gov.it/en/-/emergenza-per-covid-19-on-line-questionario-nazionale-sui-cambiamenti-delle-abitudini-alimentari, accessed on 20 May 2021), social media (e.g., Facebook and Twitter), instant messaging apps (e.g., WhatsApp), and by mailing personal contacts. Google Form^®^ was used for data collection mask creation to a self-reported compilation. This simple non-probabilistic approach to recruit respondents online, by inviting them to follow a link to a survey placed on a web page, email, or other similar means, is defined “river” sampling by [29]. The data collection was opened between the 22 April and 10 May 2020, which was the final period of the confinement in Italy. This timing permitted to have a recent recall of respondents’ behaviors during the days of restriction. 

### 2.2. The Questionnaire

The online questionnaire consisted of 63 items (Appendix A) and was developed in order to evaluate eating habits and lifestyle changes during the Italian lockdown following the 2020 spring phase of the coronavirus Sars-CoV2 pandemic. The questionnaire was designed by a multidisciplinary group that included nutritionists, public health specialists, and statisticians using literature information and considering different on-going assessments in order to avoid overlapping with other Italian surveys. The questionnaire covered three fields of interest, namely (i) usual adherence to the Mediterranean Diet before the lockdown; (ii) the assessment of dietary pattern changes during the lockdown; (iii) the assessment of lifestyle changes during the lockdown.

The first part of the survey covered sociodemographic information (gender, age, region of residence, education) and self-reported weight and height. After this, the 17-item questionnaire by the PREDIMED PLUS study [30,31,32] was used to investigate the respondents’ usual adherence to the Mediterranean Diet. This module was integrated with a specific question on water consumption. The questionnaire asked for the frequency of traditional Mediterranean food consumption (daily or weekly) defining the portion or asking for the use of food with “yes or no” options or the preference between different food options. With respect to the original PREDIMED PLUS questionnaire [32], the portion sizes were modified according to those defined by Italian nutritional recommendations, using the country specific Food-Based Dietary Guidelines [33]; the portion sizes used in this assessment are reported in Appendix A. No other changes were introduced in terms of food groups, scoring, and the calculation of adherence to the Mediterranean Diet. To evaluate changes in food consumption during the lockdown, the respondents were asked to consider whether there had been any variations in the items listed (an increase, a decrease or same as before). In addition, changes in general eating habits and lifestyle were analyzed using the Likert scale with five levels (1 Strongly disagree, 2 Disagree, 3 Neither agree nor disagree, 4 Agree, 5 Strongly agree) [34]. Specific questions were addressed on food purchases, management of leftovers (food waste prevention), meals habits (conviviality), physical activity, perceived body weight increase, and on opinions regarding vulnerable population groups, i.e., children and adolescents, if present in the family, and the elderly.

### 2.3. Statistical Analysis

Respondents transmitted the completed questionnaire to the Google platform and the final data base was downloaded as a Microsoft Excel sheet. The “river sampling” technique used in this study does not ensure that the population structure is captured in the sample composition, so a weight system based on the Italian population statistics [35] was adopted for gender, age class, and region of residence. The magnitude of the educational level bias was too big to be weighted without creating further errors. Descriptive statistics were carried out on categorial and continuous variables.

Based on the PREDIMED PLUS scoring, adherence to the Mediterranean Diet (AMD) was categorized into four categories: low (score 0–6), low to moderate (score 7–8), moderate to high (score 9–10), and high (score 11–17) [32]. To quantify changes in AMD during lockdown, the methodology of Molina-Montes et al. [36] was adapted to the data of the present assessment. Specifically, 1 point was assigned to changes in line with Mediterranean Diet recommendations, and 0 points were assigned to changes not in line with Mediterranean Diet recommendations, and 0 points were assigned for food intake that did not change. This means that 1 point was given when respondents reported an increase in the consumption of foods that contribute positively to AMD, such as olive oil, vegetables, fruit, whole grain cereals, legumes, fish and shellfish, nuts, white meat, and sofrito. Similarly, 1 point was given when a decrease was reported in the consumption of foods not characterizing the AMD (i.e., white bread, red meat, butter and margarine, sugary drinks, sweets and pastries, beverages with sugar added, non-whole wheat pasta and rice, and wine. Points were added up and based on quartiles distributions, 4 changing classes were defined: no changes (score 0), small changes (score 1–2), medium changes (score 3–4) and large changes (score > 4).

A contingency analysis was performed to respond to the research question of evaluations of potential connections between the different variables, e.g., to have the possibility to evaluate changes in food consumptions during the lockdown, analyzing various quantitative and qualitative variables at the same time, with possibility of identifying the changes with the greatest influence. This common statistical approach is based on processing double-entry tables and the assessment of chi-squared statistics. 

A multivariate data analysis was carried out to identify dietary change patterns. Specifically, the analysis was conducted in two steps. In the first step, the dimensions of the consumption changes were identified by applying the Nonlinear Principal Component Analysis (NL-PCA) [37,38,39] to the 18 ordinal variables of food group consumption variations during the lockdown, categorized at three levels, labelled as 1 = decreased consumption, 2 = same consumption as before, 3 = increased consumption, respect to usual dietary patterns in the period prior to the lockdown. The relevant dimensions were selected following Kaiser’ s rule [39]. Subsequently, individuals were classified in respect to the dimensions identified by performing a two-stage cluster analysis [39]. In the first stage, the hierarchical Ward method was adopted to identify the optimal number of groups; in the second stage, the non-hierarchical k-means algorithm was applied to define clusters [40]. Finally, the cluster profiles were arranged in tables reporting percentage in the cluster of main important items. Specifically, a chi-squared test was performed to check significant differences (*p* < 0.05) between the percentage in the cluster and the percentage in the rest of the sample.

A statistical analysis was performed using the IBM SPSS Statistics, version 25.

The overall number of people who participated in the online survey was 2878. The final sample consisted of 2768 participants. Hence, after the data cleaning, the percentage of valid questionnaires was 96%. The 110 subjects not included in the sample did not provide their consent on personal data processing and, even with the completion of the questionnaire, were excluded from the analysis.

## 3. Results 

### 3.1. Sociodemographic Variables

The participants came from all regions of Italy, with a prevalence of respondents from Northern regions (46.1%). Men and women were equally distributed (48.2% and 51.8% respectively), with ages mainly ranging from 30 to 69 (65.1%), and with high educational level (69%). The details of the sociodemographic characteristics of the sample are reported in Appendix A.

### 3.2. Adherence to the Mediterranean Diet (AMD) before the Lockdown

More than half of the sample (62%) showed lower AMD (low 34.5%; low to moderate 27.1%). Moderate to high AMD (19.1%) and high AMD (19.3%) were reported by the rest of respondents with the same proportion.

In our sample, dietary habits in line with the Mediterranean Diet recommendations were frequently observed, as far as concerning the frequency of the consumption of vegetables and nuts, whole grain cereals (i.e., bread, pasta, rice), and legumes. Sugary drink consumption was reported by one fourth of respondents, and most participants declared not using butter and margarine as added fat, preferring to use olive oil. The consumption of wine in this sample was generally very low. Eating habits different from Mediterranean Diet recommendations were reported by nearly half of the sample, who consumed red meat more than once a week and did not follow the correct consumption of whole grain cereals. In addition, adding sugar to drinks was a common habit and eating sweets and pastries was frequently reported, three or more times per week (Appendix A).

### 3.3. Changes in Eating Habits and Lifestyle during the Lockdown

To evaluate the effect of the lockdown on Italian food habits, changes in the consumption of typical Mediterranean Diet food were asked, and the results were reported in Figure 1. For several items, the dietary changes during the confinement led to a consumption pattern more in line with the Mediterranean Diet principles, given the increase in the consumption of vegetables (28.5%), fruit (24.4%), water (19.9%), legumes (22.1%), olive oil (18.9%), nuts (12%), whole grain cereals (13.1%), and fish or shellfish (14%). Other elements in line with the Mediterranean Diet principles were the reduction in consumption of red meat (22.2%), sugary drinks (16.3%), and butter and margarine (12.9%). The confinement period was also characterized by changes in consumption patterns that worsened the dietary profile of the respondents. In fact, the intake of sweets and pastries (36.9%) and wine (16%) increased. In addition, although a small percentage of respondents increased their fish and shellfish consumption, almost a quarter (23.8%) reduced the intake of these foods.

In Table 1, the results of a set of questions aimed to describe the behavioral changes during the restriction period are reported. Regarding food purchasing and cooking, 28.8% of the sample paid attention to food prices and around 50% of respondents did not change their usual cooking habits. However, for approximately one third of them, the lockdown led to an improvement in their cooking abilities and their willingness to try new foods. In this sample, an awareness of food surplus and waste was reported by nearly 80% of participants who claimed to consume all the food they cooked and reported to have had the capacity to store surplus and consume the leftovers. In terms of the perceived quality of the diet, 45.5% of respondents did not report an improvement in their eating habits, with 22.6% of people increasing their consumption of comfort food, 19.2% that increased the consumption of snacks, and 35.3% reporting the need to go on a diet to lose weight. A very high proportion of the sample (85%) did not find any difficulties in doing a separate collection of waste. Finally, 66% of the participants believed that the elderly experienced problems in buying food and suffered from isolation.

Physical activity levels and perceived body weight changes are reported in Figure 2. A sedentary lifestyle was very common with most of respondents that did not perform exercises as recommended (64.5%) (Figure 2a). This reflects the perception of an increased body weight reported by more than 30% of participants at different levels Figure 2b. This result is line with Q12 of Table 1 in which around one third of respondents declared the need to go on a diet to lose weight.

As a conviviality indicator and considering the changes in household practices because of the lockdown, breakfast habits and children’s involvement in cooking activities were specifically investigated. A large proportion of respondents did not change the breakfast habits, while 12.1% of them declared to have had more time for breakfast with whole family (Figure 3a). One third of the families reported an increased involvement of children in the kitchen activities with a specific interest in learning new things on food and nutrition (Figure 3b). These results are paired with the answers of Q15 of Table 1 in which 47% of respondents reported having increased the occasion of conviviality during the lockdown.

### 3.4. Effects of Lockdown on the Relationship between Adherence to the Mediterranean Diet (AMD) and Eating Habits and Lifestyle Changes 

Firstly, to evaluate the magnitude of food habit changes during the lockdown in relation to the usual AMD, an estimation was made summing up the number of changes in line with Mediterranean Diet recommendations. The higher the score of classes of changing, the higher the improvement in the usual dietary habits, measured with AMD. The results are shown Table 2 (Total column). For 26.4% of participants the lockdown had no impact on dietary habits (no changes, score 0), 30.6% experienced small changes (score 1–2), while 20.7% and 22.4% showed a progressive improvement in dietary habits in line with the Mediterranean Diet (moderate-score 3–4 and large changes-score > 4, respectively). Secondly, the usual AMD was compared with the dietary pattern changes during the lockdown, showing that the “no changes” group had a significantly higher proportion of low AMD (32.7%), while in the “large changes” group there was a preponderance of high AMD (30.1%) (Table 2).

A further analysis showed how the AMD groups changed food consumption (Table 3) and lifestyle (Table 4).

Statistically significant differences among high and low AMD groups were observed for the consumption of some foods either in term of intake increasing or decreasing. As reported in detail in Table 3, respondents with high AMD reported an increase in the consumption of nuts, whole grain cereals, fish or shellfish, and legumes and a considerable reduction in consumption of red meat, sweets and pastries, and sugar added to beverages, while sugary drinks intake remained unchanged. In addition, during the assessed period, the group with high AMD respect to that with low AMD, followed closely the recommendations drinking more water than usually and reducing the intake of wine with statistically significant differences. On the other hand, respondents who have a low AMD did not change consumption of whole grain cereals, and legumes, and 7.9% of them reduced the consumption of vegetables compared to respondents with the highest index. Moreover, the low AMD respondents increased the consumption of pasta and rice, and white bread, red meat, butter and margarine, sweet and pastries, and sugary drinks more than high AMD category. In addition to this, people with low AMD decreased wine consumption less than the others.

As shown in Table 4, respondents with high AMD (moderate to high, and high) improved their eating habits more than those with low AMD who also reported an increased consumption of comfort food and that they needed to go on a diet to lose weight more than respondents with high AMD.

### 3.5. Ponderal Status (BMI) and Changes in Eating Habits

Most of the assessed respondents had a BMI corresponding to normal weight (57.1%), whereas 29.1% were overweight and 10.3% obese. A very small proportion of the sample had BMI values corresponding to underweight (3.5%). Normal BMI values were frequently reported by respondents who have higher AMD (64% moderate to high and 60% high). On the other hand, low AMD resulted most common in respondents with BMI corresponding to underweight (5%), overweight (33%), and obese (15%). These data are not shown in the tables below.

The relationship between BMI classes and dietary changes during the lockdown period showed that underweight people generally maintained the same eating habits. Details of the food group consumption changes are reported in Table 5. In general, this group did not change the consumption of 14 items (olive oil, fruit and vegetables, legumes, fish and shellfish, nuts, sofrito sauce, wine, water, beverages with added sugar, white bread, red meat, butter and margarine, sugary drinks). A similar trend was seen for normal weight respondents, with 9 items (olive oil, fruit and vegetables, white bread, legumes, beverages with added sugar, sofrito sauce, non-whole pasta and rice, and water) remaining unchanged. On the other hand, overweight and obese respondents reported significant modifications in their consumption of different types of food. It was observed that overweight and obese respondents declared to have increased the use of olive oil more than underweight and normal weight respondents. Almost all underweight and normal weight respondents did not change their habits related to olive oil use and fruit and vegetables consumption. All groups reported an increase in consumption of vegetables with higher differences only for the group of obese respondents who also increased their intake of legumes and fish or shellfish. Considering foods that were less consumed, obese respondents decreased the intake of sugary drinks and wine and avoided adding sugar to beverages more than the other categories. Finally, although sweet and pastries consumption increased in all categories, there was a percentage of obese respondents which reduced the consumption of these food more than the other categories. 

### 3.6. Dietary Change Patterns Clusters during the Lockdown

The results of the NL-PCA led to consider the first 4 components as synthetic indicators of food consumption changes, explaining 47% of variance (data not shown). Cluster analysis applied to the 4 considered components led to identifying 4 groups of individuals (Table 6; Appendix A). Group 1, the “healthy eaters” and group 4, the “more eaters” included respondents who increased food consumption of all food categories (group 4) and mainly of healthy foods (except for sweets and pastries) (group 1). Group 2, the “less eaters” included respondents who decreased overall food consumption and group 3, the “usual eaters” who included those that did not change their food consumption pattern.

The 4 groups were profiled according to the socio-demographics, AMD, BMI, body weight changes, and physical activity (Table 7), and to lifestyle and eating habits (Appendix A). These groups can be further characterized. All results were statistically significant. The “healthy eaters” were mainly women and elderly, had high AMD and reported an improvement in eating habits. A large proportion of them shared their meals with the whole family (49.4%) and reported difficulties in finding food they needed. Even with predominantly favorable dietary changes, these respondents declared that they had eaten more comfort food, gained weight (3–5 kg), and had been sedentary.

The “less eaters” group included mainly men (66.3%), and respondents with polarized age, including young (18–29 years) (21.5%), and old people (50–69 years) (47.7%). The “less eaters” had low to moderate AMD (41.9%) and reduced physical activity (16.6%, less frequent). Moreover, most of this group respondents found difficulties in finding food that they needed (34.9%) and paid attention to prices (35.7%). A positive characteristic of the “less eaters” group is having shared the meal with the whole family (52.9%). 

In the “usual eaters” group a high prevalence of women (54.9%) and elderly (26.2%) was observed. The AMD of the “usual eaters” was lower (38.6%) with respect to the other groups. The respondents included in this group declared that they did not need to go on a diet (50%), and a high prevalence of them were sedentary (39.1%). The attitude of not changing the dietary intake of the “usual eaters” was coupled with the similar trend in their habits resulting in not having cooked better (29.4%), not having improved their diet (52.8%), and having always had the same cooking habits (53.6%).

In the “more eaters” group there was a preponderance of women (53.1%) and people aged 18–29 (20.3%) and 30–49 years (44.2%). These respondents increased their intake of comfort food (25.7%) and snacks (24.6%) and reported the intention to go on a diet (44.6%), having declared to performing physical activity at an insufficient level (less than 1 or 2 times per week) (14.8%). In addition, even though these respondents found difficulties in finding food (30.5%), they claimed to have improved their cooking (46.9%), to have tried new foods (51.7%), and changed their cooking (33.9%). The “more eaters” claimed to have improved their general eating habits (36%) and reported sharing the meal with the rest of the family more than the others (63%).

Respondent behaviors concerning food waste was similar in all groups. No differences were found in the assessment related to the ability to store food correctly, and to perform separate collection of food waste, both practices were shared by around 80% of each group respondents. In addition, most of them (around 70%) declared to have consumed all food stored included the leftovers, however, the “healthy eaters” were more keen than other groups in adopting these practices (83.1%).

## 4. Discussion

This paper describes the changes in eating habits and lifestyle that occurred during the period of the containment measures which were adopted to limit the impact of the spread of coronavirus Sars-Cov-2 in Italy during spring 2020. The objective of the assessment was to provide a picture of a unique situation, and to evaluate if, and to what extent, consumers, were following the dietary recommendations that experts and institutions provided. This goal was largely accomplished, considering that the present paper has aspects of similarity with other assessments carried out in the same period but also provides new and complementary information on the topic since the lockdown started. In fact, the findings resulting from the data analysis and pooled variables presented here provided additional information and an original vision of the effect of the lockdown on dietary behaviors and lifestyle changes. 

An important finding that comes from this assessment is that a large prevalence of the participants, during the lockdown period, seems to have followed the dietary recommendations, increasing their consumption of healthy foods such as legumes, fruit and vegetables and whole grain cereals. Thus, the present data do not confirm the worsening of the dietary profile as stated by Ammar et al. [41] in a survey covering Europe, North-Africa, Western Asia, and the Americas carried out during the isolation period. In fact, in Italy, the scenario was different considering that other studies reported an increased adherence to the Mediterranean Diet and a highest intake of healthy foods during the confinement [3] as well as most sustainable food choices with increased consumption of organic and locally grown food [42]. It is interesting to point out that another Mediterranean country, Spain [43], also reported the adoption of healthier dietary habits and increased adherence to the Mediterranean Diet during the confinement period. 

Commonly, when dietary changes occur, improvements in the quality of the diet can be counteracted by unhealthy eating behaviors. As observed in other studies [41,44], in our sample, an increase in consumption of sweets, pastries, comfort food, and red meat was observed, alongside with a decrease in the consumption of fish and shellfish, as modifications of dietary habits not coherent with the recommendations. The changes observed in the present sample reflected the national trend of food purchasing reported by the Service Institute for the Agri-Food Market (ISMEA) study carried out in Italy during March 2020 showing that, during the first period of the lockdown, there was an increased demand for canned goods, comfort food, and a decrease in the sale of fresh products [22]. On the other hand, the improved eating habits and cooking skills together with the willingness to try new foods and recipes reported by many of our respondents correspond to another research carried out by the same institution that continued to monitor the agri-food purchases system during the pandemic outbreak. In this further study carried out in April 2020, ISMEA highlighted that after an initial period in which consumers bought long-life products, they then changed their purchasing habits mainly towards ingredients for cooking, such as flour, yeast, sugar, eggs, and butter [23]. These results confirmed that consumers became more careful about the quality of food they were cooking, also reducing the purchasing of ready-to-use foods. The fact that our results are in line with these findings represents a methodological reinforcement of the outcomes we obtained, given that, despite using a non-probabilistic sample, we observed the same tendencies showed by a national representative assessment.

In brief, we observed an improvement in the quality of the diet considering the reported increased consumption of healthy foods and the reduction in consumption of some foods not in line with the Mediterranean Diet. However, the increase of the frequency of consumption of certain food categories (e.g., olive oil) and the increased consumption of sweets, pastries, and comfort foods coupled with physical activity reduction, could be claimed to be responsible for the perceived increased body weight. The sedentary lifestyle and the weight gain reported in this study are confirmed by others similar studies performed in Italy [3,42,45]. In addition to these, our work is in line with the conclusions of a recent scoping review carried out by Bennett et al. [46] which reported that the lockdown had both positive and negative impact on dietary practices across Europe and worldwide, and negative eating habits were associated with other lifestyle outcomes as weight gain and a reduced physical activity. Moreover, findings on physical activity, weight gain, and some dietary changes (sweets, vegetables and fish) are in line with the results of a study carried out in Austria, Poland, and the United Kingdom by Skotnicka et al. [47].

More than a half of respondents of this survey has low adherence to Mediterranean Diet. In other surveys carried out in Italy or in other Mediterranean countries during the pandemic, the Mediterranean Index was used mainly to evaluate the changes related to lockdown, hence it is difficult to make comparison. However, the proportion of respondents of our sample with low adherence to Mediterranean Diet deserves attention. These results need to be confirmed by other assessments with a sampling that allows a generalization of the outcome. However, in Bonaccio et al.’s [48] study that surveyed a Southern Italy large cohort (113.262 subjects), higher income and education were independently associated with a greater adherence to Mediterranean Diet. Among our sample subjects, the high educational level was overrepresented, so we could speculate that in a comprehensive sample including a greater proportion of low educational level, the low adherence to Mediterranean Diet could be higher than that observed in the present study.

A further novelty of this assessment with respect to others published on the same topic, was the cluster analysis that provides an overview on the effects of lockdown on subjects studied and how food consumption changes were related with other variables. The confinement polarized behaviors, with people who already had a high AMD increased their consumption of healthy food (the “healthy eaters”). Consumers with good eating habits were also the keenest to follow the recommendations during the lockdown. On the contrary, respondents that did not change their consumption pattern, the “usual eaters” were those having usual low adherence to the Mediterranean Diet. For this category the lockdown exacerbated a situation that was already unfavorable [44]. However, an increase in the consumption of comfort foods was observed in all groups as a possible compensation for the negative emotions caused by ongoing emergency and the bad news coming from the media or, simply, as consequence of the boredom of staying at home [49,50]. 

As mentioned, we observed an increase in body weight and sedentary lifestyle. These findings could be interpreted in the light of the results from the cluster analysis showing that low levels of physical activity were reported frequently by all groups regardless of their consumption patterns. Moreover, the cluster analysis also highlighted many of socio-economic aspects of the lockdown that deserve comments. The “less eaters” group paid attention to food prices and found difficulties in buying food. These results may be a consequence of the lack and/or loss of job that impacted on the youngest [51,52], and the isolation which affected the oldest [53,54], considering the high prevalence of these population groups among the “less eaters”. Related to this, the results concerning the elderly showed that even though there was a percentage of this population group that followed MD recommendations, they had also the attitude to not change their eating habits and not improve their food consumption, confirming the fact that this population group often needs nutritional intervention to improve the quality of their diet. Children were to some extent involved in the kitchen and food preparation activities during lockdown. Targeting this population group in educational campaigns in this period was an important initiative [55] also considering the high level of childhood obesity in Italy [56] and the worsening of this situation globally due to the pandemic [45,57].

To conclude the analysis, two other aspects of lifestyle should be noted. Firstly, the habit of eating together increased positively among most of the respondents (groups 1 “healthy eaters”, group 3 “usual eaters” and group 4 “more eaters”). Secondly, all the participants to this study were, without relevant differences between the 4 groups, very sensitive to the problem of food waste. These findings are promising since they may be a signal of the increased attention of Italian consumers towards the negative consequences of throwing away food. Other studies confirmed the attention of Italian consumers to food waste both during the lockdown [58,59] and before the pandemic [60].

In addition to consumers’ behavior changes, the coronavirus crisis is causing many problems and is worsening pre-existing conditions. Many people around the world have lost their jobs and inequalities related to wealth, income level and social protection have exacerbated [61,62,63]. Moreover, food insecurity [64] increased among who have already suffered from poverty in developing countries [65] but disparities came out also in developed countries [66,67]. The suggestion that comes from this study is that the food system in Italy at the household level during the first lockdown was an important element to maintain a daily acceptable routine for the whole family in a pandemic situation which had relevant socio-economic impacts. This situation should be the starting point to reorganize the system during the recovery and to be more aware of how to handle a possible similar future crisis.

A limitation of this work is related to the bias of the “river sampling”, as far as possible limited using weighted data that corrected the sociodemographic structure of the sample since Italian population statistics [35]. The “river sampling” has advantages and disadvantages. The facility to reach a large number of participants at limited costs widespread the application of this method, especially during the lockdown period. This tendency is increasing independently from the restrictions of the pandemic. In fact, despite limitations, non-probabilistic online surveys are now frequently used to make claims about the general population in social science and policy research. According to Lehdonvirta et al. [29], this kind of sampling can be used to describe certain phenomena and their safest use is in examining members of the sample itself. In addition, this sampling could be used to make tentative inferences, but the effect sizes would not be accurate, and could be affected by the selection biases. We perform the analysis of the data in light of these methodological inputs treating the sample as is and considering possible bias. For example, the high educational level of the participants of this survey, could partially explain the increased quality of the dietary patterns observed in our and other opportunistic samples selected with similar methodology. This is a limitation of the studies carried out during the lockdown, but it is also true that to accomplish the objective of providing a picture of that specific moment no more other options were possible, considering the restrictions and, above all, the need to perform the data collection without the appropriate planning. Other limits of this study consist of the self-reported answering that could affect the reliability of the responses and the fact that eating habits were based on participants’ perception of food intake that may not reflect true intake during lockdown. However, the large sample size and the confirmation of our results with other similar survey support the quality of the data collected.

## 5. Conclusions

The results of this study together with the other assessments carried out in the same period and in different population groups, showed that the lockdown in Italy improved the quality of diet but had negative effects both on the quantity of food consumed and on the levels of physical activity. This situation led to a general increase in body weight and to an exacerbation of sedentary lifestyle that was already common in Italy. These conditions should be carefully considered as detrimental changes with potential significant impact across the population groups, in particular, for the most vulnerable, such as the elderly. Nevertheless, the involvement of children in kitchen activities, the increased occasions of conviviality, and the attention on the issue of food waste should be considered important positive habits to highlight.

This survey should be considered an important element that allows to consider nutritional aspects in the framework of the pandemic approach, given that the pandemic is still on-going and because also in 2021 people experienced different levels of confinements that exposed them to the same detrimental conditions. An additional value of this assessment is the timing. Having distributed the survey until nearly the end of the first lockdown period, the data collection provided a comprehensive overview about how consumers adapted their habits to this unpredictable situation.

Even though the acute phases of Coronavirus disease 2019 are increasingly under control thanks to the ongoing vaccination campaigns that is reducing deaths and severe cases, the overall consequences of the pandemic period would significantly impact on people nutritional status as shown in this paper and at larger extent on the health system. The increase in unhealthy dietary habits and deterioration of nutritional status found in this study may contribute to a general decline of health among population groups, considering that being overweight and obese makes individuals more susceptible to chronic health conditions and diseases.

Future public health actions should consider the worsening of people’s nutritional status one among other long-term health consequence of the pandemic.

## Figures and Tables

**Figure 1 nutrients-13-02279-f001:**
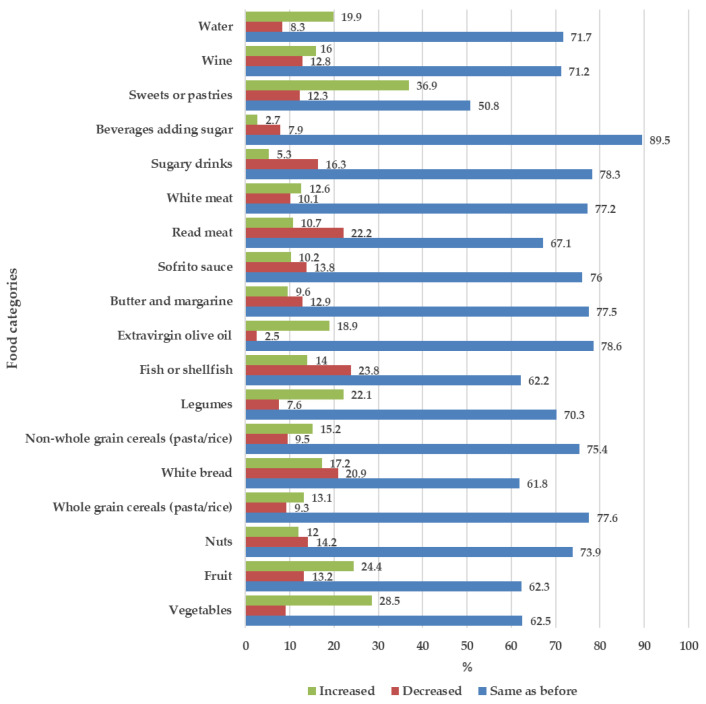
Food consumption changes during the lockdown.

**Figure 2 nutrients-13-02279-f002:**
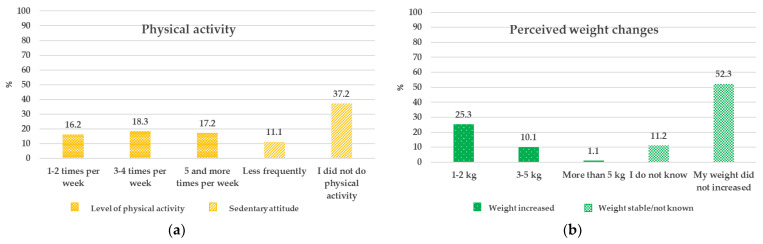
The effect of lockdown on body weight: (**a**) Physical activity; (**b**) Perceived weight changes.

**Figure 3 nutrients-13-02279-f003:**
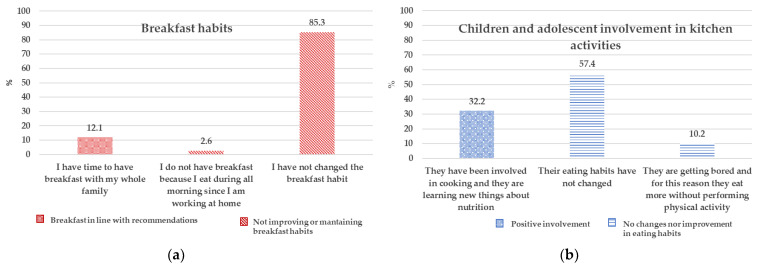
Domestic practices during lockdown: (**a**) Breakfast habits; (**b**) Children and adolescent involvement in kitchen activities.

**Table 1 nutrients-13-02279-t001:** Respondents’ eating habits and lifestyle changes during the lockdown.

Eating Habits and Lifestyle Changes	Strongly Disagree (%)	Disagree (%)	Neither Agree Nor Disagree (%)	Agree (%)	Strongly Agree (%)
Q1. I need to pay attention to my spending, and I should limit my purchases of expensive foods	25.4	15.4	30.4	19.4	9.4
Q2. My cooking habits did not change, I continued to cook as before	11.7	15	25.3	22.9	25.2
Q3. I feel I am better at cooking	17.4	9.6	37.4	22.1	13.4
Q4. I have purchased and tried new foods that I never tasted before	24.9	15.2	26.7	23.5	9.7
Q5. I cannot always find the foods I would like to eat	33.6	20.6	17.6	21	7.3
Q6 I ate all food I had cooked, including leftovers	7.1	5.1	11.4	26.1	50.3
Q7. I do not know how to store and consume all foods that I bought	62	19	11.6	5.5	1.9
Q8. Separate collection of waste is very difficult, I cannot do it	70	15	8.1	3.5	3.6
Q9. I eat more comfort food than before (i.e., prosecco, snacks, sweets, etc)	48.8	17.3	11.3	14.5	8.1
Q10. I eat a lot of snacks during the day	42.1	22.1	16.6	13.3	5.9
Q11. I have improved my eating habits	26.6	18.9	31.8	14.6	8.1
Q12. I need to go on a diet to lose weight	27.2	18.1	19.3	18	17.3
Q14. I have the perception that the elderly have difficulties with shopping and suffer from social isolation	8.3	8.4	17.2	27.3	38.7
Q15. I eat the main meals together with the rest of my family more frequently than before	18.7	7	27.3	16.4	30.6

**Table 2 nutrients-13-02279-t002:** Relation between food habits’ changes quartiles during the lockdown and AMD classes before the lockdown (percentage).

Changes of Food Habits during Lockdown (Classes–Quartiles)	Total	AMD Classes before Lockdown			
		Low	Low to Moderate	Moderate to High	High
No Changes	26.4	32.7 ^a^	22.9	24.8	21.6
Small Changes (1–2)	30.6	27.2	33.8	28.3	34.2 ^b^
Medium Changes (3–4)	20.7	23.2 ^c^	19.3	25 ^d^	14.1
Large Changes > 4	22.4	17	24 ^e^	21.9 ^f^	30.1 ^g^
Total	100	100	100	100	100

Significance test results based on pairwise tests: ^a^ Low vs. Low to moderate, *p* = 0.000; Low vs. Moderate to high *p* = 0.009; Low vs. High *p* = 0.000; ^b^ High vs. Low, *p* = 0.25; ^c^ Low vs. High, *p* = 0.000; ^d^ Moderate to high vs. High, *p* = 0.000; ^e^ Low to moderate vs. Low, *p* = 0.002; ^f^ Moderate to high vs. Low, *p* = 0.019; ^g^ High vs. Low, *p* = 0.000; High vs. Moderate to High, *p* = 0.014.

**Table 3 nutrients-13-02279-t003:** Food categories consumption during the lockdown by the AMD before the lockdown (*percentage*).
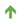
 or 
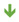
 describes favourable food consumption changes, = describes unchanged food consumption, 
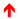
 and 
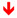
 describes unfavourable food consumption changes; ** p* < 0.05.

Food Categories	Food Consumption	Total	AMD
Low	Low to Moderate	Moderate to High	High
Nuts	Increased	12	7.8	* 13.6	* 14.5	( 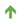 ) * 14.6
Decreased	14.2	15.1	15.5	17.1	7.7
Same as before	72.8	77.1	70.9	68.4	77.7
Legumes	Increased	22.1	17.1	* 22.5	* 26.3	( 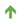 ) * 26.6
Decreased	7.6	7.8	8.3	7.3	6.5
Same as before	70.3	(=) * 75.1	69.2	66.4	66.9
Whole cereals	Increased	13.1	9.2	11.4	* 15.6	( 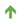 ) * 20.1
Decreased	9.3	7.6	10.6	10.3	9.5
Same as before	77.6	(=) * 83.2	78	74.2	70.4
Vegetables	Increased	28.5	28.1	28.9	27.2	29.7
Decreased	9	( 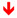 ) * 7.9	* 14.3	* 9.5	3
Same as before	62.5	64	56.8	63.3	67.3
Fish and shellfish	Increased	14	12.9	11	* 15.5	( 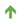 ) * 18.7
Decreased	23.8	22.4	28.9	26.5	16.7
Same as before	62.2	64.7	60.1	58	64.6
White bread	Increased	17.2	( 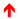 ) * 20.3	* 21.4	* 15.2	8
Decreased	20.9	16.2	19.7	18.4	33.6
Same as before	61.9	63.5	59	66.4	58.4
Non-whole pasta and rice	Increased	15.2	( 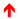 ) * 17.3	* 16	* 17	8.4
Decreased	8.5	5.8	9.1	10.6	* 15.3
Same as before	76.3	76.9	74.8	72.4	76.3
Red meat	Increased	10.7	( 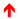 ) 13.7	9.2	* 12.3	5.9
Decreased	22.2	19	19.3	19.8	( 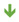 ) * 34.1
Same as before	67.1	* 67.3	* 71.6	* 67.9	59.9
Butter and margarine	Increased	5.3	( 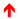 ) * 12.7	9.8	8.4	4.8
Decreased	16.3	12.2	13.7	11.1	14.5
Same as before	78.4	75.1	76.4	80.5	80.7
Sweet and pastries	Increased	36.9	( 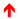 ) * 43.5	34.6	* 42	23.3
Decreased	12.3	6.6	* 15.5	* 12.9	( 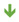 ) * 17.3
Same as before	50.8	49.9	49.9	45.1	* 59.4
Drinking beverages adding sugar	Increased	2.7	3.3	3.1	2.5	1.1
Decreased	7.9	* 5.3	* 11.3	5.3	( 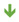 ) * 10.1
Same as before	89.4	* 91.4	85.5	* 92.3	88.8
Sugary drinks	Increased	5.3	( 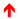 ) * 8.8	4.4	3.8	2
Decreased	16.3	14.7	* 21.9	13.9	14
Same as before	78.4	76.5	73.8	* 82.3	(=) * 83.9
Wine	Increased	16	16.5	15	19.8	12.6
Decreased	12.8	( 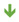 ) * 8	* 16	* 14.5	( 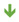 ) * 15.2
Same as before	71.2	75.5	69	65.6	72.2
Water	Increased	19.9	17	20.2	19.5	( 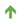 ) * 25.3
Decreased	8.3	8.2	7.6	10.1	7.8
Same as before	71.7	* 74.8	72.2	70.5	66.9
Total		100	100	100	100	100

**Table 4 nutrients-13-02279-t004:** Eating habits and lifestyle during the lockdown by the AMD before the lockdown (*percentage*). 
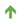
 describes favourable food consumption changes, 
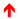
 and 
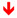
 describe unfavourable food consumption changes; ** p* < 0.05.

Eating Habits and Lifestyle		Total	AMD
Low	Low to Moderate	Moderate to High	High
I have improved my eating habits	Low	45.5	( 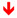 ) * 54.2	* 40.6	* 44.1	38.5
Medium	31.8	27.7	36.7	31.1	33.1
High	21.7	18.1	22.8	* 24.7	( 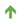 ) * 28.4
I eat more comfort food than before	Low	66.1	53	71.7	69.1	* 78.5
Medium	11.3	* 17.1	9.6	7.9	6.9
High	22.6	( 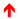 ) * 29.9	18.7	23	14.6
I need to go on a diet to lose weight	Low	45.3	79.6	77.6	* 85.6	* 83.5
Medium	19.3	18.9	20	17.3	21
High	35.4	( 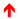 ) * 38.1	* 36.7	* 36.8	27.1
Total		100	100	100	100	100

**Table 5 nutrients-13-02279-t005:** Food categories consumption during the lockdown by BMI class before the lockdown (*percentage*). 
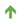
 or 
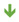
 describe favourable food consumption changes, = describes food consumption unchanged; * *p* < 0.05.

Food Categories	Food Consumption	Total	BMI Class
Underweight3.5	Normal Weight57.1	Overweight29.1	Obese10.3
Extra virgin olive oil	Increased	19	10.6	16.1	( 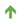 ) * 23.5	( 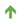 ) * 25
Decreased	2.5	0.6	2.6	2.9	1.2
Same as before	78.5	* 88.8	81.3	73.6	73.8
Fruit	Increased	24.5	19.7	23.4	27.3	24.5
Decreased	13.3	4.6	9.6	* 17.6	* 24.7
Same as before	62.2	(=) * 75.7	(=) * 67.1	55.1	50.7
Vegetables	Increased	28.6	20.9	27.9	28.1	( 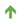 ) * 36.3
Decreased	9	4	7.5	11.9	11
Same as before	62.4	(=) * 75	(=) * 64.6	60	52.7
Legumes	Increased	22.2	16.6	22.3	19.5	( 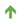 ) * 31
Decreased	7.6	3.8	8	6.3	10.4
Same as before	58.6	* 79.5	* 69.7	* 74.2	58.6
Fish and shellfish	Increased	14.1	5.8	14.4	11.2	( 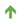 ) * 23.1
Decreased	23.9	12.1	23.7	27.4	19.5
Same as before	62	* 82.2	61.9	61.5	57.3
Nuts	Increased	12	5.4	12.2	11.8	13.3
Decreased	14.2	8.8	15.5	12.7	12.9
Same as before	73.8	* 85.8	72.3	75.4	73.8
Butter and margarine	Increased	9.6	3.7	10.3	8.2	11.8
Decreased	12.9	4.3	11.6	* 15.8	* 14.9
Same as before	77.5	* 92	78	76	73.3
Red meat	Increased	10.7	4.8	10	12.8	10.6
Decreased	22.2	12.3	23.4	20.1	25
Same as before	67.1	* 82.9	66.6	67.1	64.4
Sofrito	Increased	10.2	5.9	9.8	10.6	13.4
Decreased	13.9	5.7	10.1	* 16.6	* 29.9
Same as before	75.9	* 88.4	* 80.1	72.8	56.7
Sugary drinks	Increased	5.4	1	5.3	4.9	8.1
Decreased	6.4	6.73	12.8	19.3	( 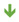 ) * 31.4
Same as before	78.2	* 92.3	* 81.9	75.7	60.55
Sweets and pastries	Increased	37	* 54.3	37.7	35.6	31.28
Decreased	12.3	7.7	10.6	13.4	( 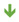 ) * 20.02
Same as before	50.7	38	51.7	51	48.7
Drinking beverages adding sugar	Increased	2.7	4.5	2.2	2.1	6.2
Decreased	7.9	6.8	6.7	4.9	( 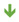 ) * 22.9
Same as before	89.4	88.8	91.1	92.9	70.8
Water	Increased	19.9	10.7	20.2	15.8	32.8
Decreased	8.4	3.9	9.2	8.1	6.2
Same as before	71.8	* 85.4	70.6	* 76.1	61.1
Wine	Increased	16	6.4	16.9	17.4	10.6
Decreased	12.8	5.7	11.4	12.4	( 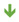 ) * 24.2
Same as before	71.2	87.9	71.7	70.2	65.2
Total		100	100	100	100	100

**Table 6 nutrients-13-02279-t006:** Dietary change patterns during the lockdown: groups distribution and characterization (*percentage*).

Groups	%	Food Consumption Pattern
Group 1- ”healthy eaters”	26.8	increased consumption of legumes, whole grain cereals and non-whole pasta and rice, nuts, sweets and pastries
Group 2- “less eaters”	7.5	decreased consumption of olive oil, fish and shellfish, nuts, legumes, white meat, whole grain cereals and non-whole pasta and rice, vegetables, red meat, butter and margarine, sugary drinks, sweets and pastries and sofrito sauce
Group 3- “usual eaters”	51.4	unchanged consumption
Group 4- ”more eaters”	41.4	increased consumption of almost all categories (coexist an increased and a decreased consumption of olive oil, whole grain cereals and adding sugar to beverages)

**Table 7 nutrients-13-02279-t007:** Cluster analysis: the relation between the 4 groups and sociodemographic variables, AMD, BMI class, body weight changes, and physical activity; * *p* < 0.05.

Sociodemographic Variables		Total	Group 1	Group 2	Group 3	Group 4
			Healthy Eaters	Less Eaters	Usual Eaters	More Eaters
Gender	Man	48.2	49.6	* 66.3	45.1	46.9
	Woman	51.8	* 50.4	33.7	* 54.9	* 53.1
Age	18–29	14.6	13	* 21.5	12.8	* 20.3
	30–49	32.2	28.7	28.8	31.2	* 44.2
	50–69	32.9	33.3	* 47.7	29.9	35.5
	≥70	20.3	* 25	2	* 26.2	0
AMD		Total	Group 1	Group 2	Group 3	Group 4
Low		34.5	30.2	25	* 38.6	33
Low to moderate		27.1	22.3	*41.9	27.2	27.5
Moderate to high		19.1	17.5	18.6	19.1	22.1
High		19.3	* 29.9	14.5	15.1	17.3
BMI class		Total	Group 1	Group 2	Group 3	Group 4
Underweight		3.5	1.8	1.7	* 4.8	2.7
Normal weight		57.2	57.1	53.4	56.9	60.4
Overweight		29.1	30	25.5	29.5	27.5
Obese		10.3	11.2	* 19.4	8.7	9.4
Body weight changes		Total	Group 1	Group 2	Group 3	Group 4
I do not know		11.2	10.1	* 18.7	10.6	11.6
My weight did not increase		52.3	51.3	46.7	* 55.8	44.3
1–2 kg		25.3	25.8	23.5	24.1	29.5
3–5 kg		10.1	* 12.1	9.7	8.5	12.3
More than 5 kg		1.1	0.8	1.5	0.9	2.3
Physical activity		Total	Group 1	Group 2	Group 3	Group 4
I did not do physical activity		37.2	* 41.1	26.1	39.1	29
Less frequently		11.1	10.9	* 16.6	9.5	* 14.8
1–2 times per week		16.2	17.2	20.8	14.1	19.4
3–4 times per week		18.3	17.5	18.2	17.9	21.6
5 and more times per week		17.2	13.4	18.3	* 19.5	15.3

## Data Availability

The archived data and all the elaboration and analysis generated and used for presentation of results in this study are fully available on request from the corresponding author.

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
