# Peer review of "Eating Habits during the COVID-19 Lockdown in Italy: The Nutritional and Lifestyle Side Effects of the Pandemic"

_nutrients, 2021, doi:10.3390/nu13072279_

Round 1

Reviewer 1 Report

This is an interesting study, describing the impact of the current pandemic on the health and wellbeing of people. I have a few comments for the authors' consideration:

Abstract - Provide some figures of important findings in the abstract. Include sample size. 

Introduction - well-written

Methods

  • Cross-sectional studies are not timeline-related. Hence it is not possible to call this a cross-sectional retrospective. 
  • Disagree that the study does not require ethical clearance. Any studies conducted among humans/animals would require ethical clearance, unless if we are talking about secondary data. 
  • line 103 - self-reported weight/height?

Results

  • Figure 1 does not add value to the paper and can be omitted. Info can be just presented in the text. 
  • Table 4 - provide p-value within the table
  • Consider moving the figures/tables that are descriptive to the supplementary doc. Keep the space within the Result section for inferential analysis. 
  • Predictive analysis will add more value to this paper. 

Discussion / Conclusion - No further comments

Author Response

See attached files

Reviewer 2 Report

This is a well-written study which  examined 17 dietary pattern changes identified by a cluster analysis that provided interesting results. The study may provide a useful contribution to the  future  nutritional recommendations in the context that many countries around the world are under third wave of pandemic now. However, the results of this study was based on participant's perception on their food intake (increase, decrease or same as before), not really reflect their true intake during lockdown. I suggest authors consider to mention it in the limitation. 

What is the response rate of this survey as authors did not report in the Method section?

Line 421: The authors stated that they observed an increase in the quality of the diet but also an increase in the quantity of food that was consumed. This sentence is not clear to me as seen in Table 7, most of the respondents answer "same as before" for the food intake they consume. 

Reviewer 3 Report

The authors of this manuscript describe the results of a cross-sectional study conducted on a convenience sample of adults in Italy during lockdown for the COVID-19 pandemic in spring 2020, focused on changes in eating habits and physical activity during that period.

The study design is adequate and as most studies conducted during movement restrictions due to the pandemic collected the information through the Internet.

The manuscript, however, has serious problems, both in the methods and the results sections, that the authors should address. In addition, the English language needs carefull reading and rewriting. Some of the issues raised could be related to language problems.

1.- The authors used a modified version of the validated PREDIMED-PLUS questionnaire to assess adherence to Mediterranean diet. In addition, they included questions about changes in eating habits, physical activity and lifestyles. Did they used validated questionnaires? Previously used questionnaires? Did they conducted a pre-test?

Were the questions exactly formulated as shown on table S1? Was any additional information provided to participants?

Was the structure of the questionnaire as shown on table 1S?

Why did you ask twice about frequency of consumption of whole pasta an rice?

What do you mean by "My cooking has not changed, I follow the same recipe"? What recipe? For cooking pasta?

2.- Did you ask individuals to measure themselves or to report their weight and height? If so, it would be more appropriate to describe it as collecting information about self-reported body weight and height, instead of anthropometric measurements.

3.- The section about data analysis is particularly problematic. the procedure followed to compute scores of change on lines 130 to 136 is not clear.

What do you mean by "1 point to each increase" and 1 point to each decrease"?

Did you compute an overall diet change score?
Did you compute a healthier diet change score and an unhealthier diet change score?

This procedure should be clearly described as it is pivotal for the analysis conducted and the results presented.

Information on line 138 is vague and unspecific. 'A contingency analysis has been performed to evaluate potential connections between the different variables'. Which was the research question?

Then, the authors describe they conducted principal components analysis and cluster analysis, but they do not describe which variables they used for the analysis, which criteria they applied, etc.

How many components did you identify? Were those patterns meaningful?

Which variables did you use in cluster analysis? Which criteria did you apply in cluster analysis?

As described, apparently the authors conducted principal components analysis to identify patterns of dietary change and then cluster analysis to identify again patterns of dietary change? Why?

3 The results section is too descriptive. All the data on figure 1 are described in the text. Please, highlight the main results in the text and use the figures and tables for detailed information.
What do you mean on lines 167-168: ' In our sample, habits coherent with the Mediterranean model were consumption of vegetables (60.7% two or more servings per day) and nuts (25.5% three or more servings per week)'.

Please further ellaborate figures 3 and 4.

What do you mean on table 4? what kind of dietary changes are shown? any change?

Table S3 will be informative as a main table, instead of including it as supplementary material. The authors can include this table and move some of the previous descriptive material to supplementary information.

Round 2

Reviewer 1 Report

I'm happy with the changes and have no further requests for amendments.